

# Radix Stellariae extract prevents high-fat-diet-induced obesity in C57BL/6 mice by accelerating energy metabolism

Yin Li, Xin Liu, Yu Fan, Baican Yang and Cheng Huang

School of Pharmacy, Shanghai University of Traditional Chinese Medicine, Shanghai, China

## ABSTRACT

*Stellaria dichotoma L.* is widely distributed in Ningxia and surrounding areas in north-western China. Its root, Radix Stellariae (RS), has been used in herbal formulae for treating asthenic-fever, infection, malaria, dyspepsia in children and several other symptoms. This study investigated whether the RS extract (RSE) alleviates metabolic disorders. The results indicated that RSE significantly inhibited body weight gain in high-fat (HF)-diet-fed C57BL/6 mice, reduced fasting glucose levels, and improved insulin tolerance. Moreover, RSE increased the body temperature of the mice and the expression of uncoupling proteins and peroxisome proliferator-activated receptors in the white adipose tissue. Thus, RSE alleviated metabolic disorders in HF-diet-fed C57BL/6 mice by potentially activating UCP and PPAR signaling.

## INTRODUCTION

Metabolic syndrome (MS) is prevalent world-wide, particularly in Western countries. It is characterized by obesity, insulin resistance, hyperlipidemia, type 2 diabetes mellitus, hypertension, and atherosclerotic cardiovascular disease (*Morikawa et al., 2004*; *Eckel et al., 2010*). Excessive calorie intake and lack of exercise are the two main reasons leading to MS. Recent relevant experimental and clinical research results can be summarized as follows: (1) develop practical methods to address the main causes of MS, and (2) identify a direct method to eliminate adverse factors, such as insulin resistance, hyperlipidemia, obesity, and hypertension (*Ginsberg, 2003*). Thus, although difficult, novel therapeutics to prevent and treat obesity are urgently required (*Apovian et al., 2015*).

Brown adipose tissue (BAT) is essential for thermogenesis and body temperature maintenance (*Harms & Seale, 2013*). When activated, BAT can express uncoupling protein1(UCP1) to release energy in the form of heat by uncoupling the protons generated by substrate oxidation during adenosine triphosphate (ATP) production (*Izzi-Engbeaya et al., 2015*). Moreover, white adipose tissue (WAT) can be used as an index of energy metabolism for it browning. UCP1, UCP2, and UCP3 are related to energy metabolism; in particular UCP1 plays a critical role in releasing electrons rather than storing them, resulting in heat release (*Kim & Plutzky, 2016*).

Corresponding authors
Baican Yang, bcy2002@sina.com
Cheng Huang,
chuang@shutcm.edu.cn

Peroxisome proliferator-activated receptors (PPARs) are members of the nuclear receptor family (*Francis et al., 2003*). PPAR α, PPAR β, and PPAR γ are their three isoforms. PPARs are crucial regulators of lipid, glucose and tissue metabolism as well as cell differentiation and proliferation, apoptosis, and host immunity (*Desvergne & Wahli, 1999*). PPARs bind to the retinoid X receptors to form heterodimers, which regulate downstream gene expression by interacting with PPAR response elements in these genes (*Wahli, Braissant & Desvergne, 1995*). PPAR α is present in the liver (*Jia et al., 2003*), heart (*Barger & Kelly, 2000*; *Gilde & Bilsen, 2003*), skeletal muscle, BAT and kidneys. It mainly mediates the uptake and β-oxidation of fatty acids in the liver and heart (*Bishop-Bailey, 2000*; *Puddu, Puddu & Muscari, 2003*). The activation of PPAR α is an effective therapy for hyperglyceridemia. PPAR γ is expressed abundantly in adipocytes, particularly in WAT, as well as in the gastrointestinal tract and macrophages (*Thompson, 2007*). It plays a key role in adipocyte differentiation, lipid accumulation, and insulin sensitivity (*Spiegelman, 1998*; *Francis et al., 2003*), and is involved in whole-body glucose homeostasis (*Barroso et al., 1999*). As we all know, PPAR γ is the target of the insulin-sensitizing agent rosiglitazone.

Radix Stellariae (RS), also called Yinchaihu, the root of Stellaria dichotoma, is a common Chinese herbal medicine used clinically to treat fever and infantile malnutrition. RS was first described in "Ben Cao Gang Mu" 400 years ago (*Teng, 1985*). According to the clinical studies, RS also has several other pharmacological functions, including anti-inflammatory (*Chen et al., 2010*), anti-cancer, and anti-allergic activities (*Morikawa et al., 2004*; *Sun et al., 2004*) as well as dilation of blood vessels (*Morita et al., 2005*). Recently, it has been reported that RS had a higher content of α-spinasterol, which has anti-inflammatory and antipyretic effect and β-carboline alkaloids in RSE with anti-allergy properties through the mice anti-allergic reaction experiment (*Morikawa et al., 2004*). Besides, new cyclicpeptids extracted from RS has been demonstrated with antitumor activity *in vitro* and mild dilation of blood vessels. Nevertheless, the effects of RS on metabolic disorders have not been reported.

In this study, we observed the effects of RSE on HF-diet-induced obesity to assay whether RS could alleviate metabolic disorders. We found that it can alleviate MS by reducing body weight and blood glucose levels, increasing insulin sensitivity in HF-diet-induced obese mice.

## MATERIALS AND METHODS

### Preparation of Radix Stellariae extract

RS was purchased from the Ningxia province. In a spherical extractor, 4L of 95% ethanol was added to 500 g of RS. Extraction was performed for 2 h at 85 °C, followed by cooling and filtering of the extract. Extraction was repeated using 50% ethanol. The extracted solutions were combined for rotary evaporation at 60 °C, under reduced pressure, till the taste of alcohol was undetectable. Finally, the concentrated solution was freeze-dried and stored at −20 °C.

## Liquid chromatograph-high resolution mass spectrometry

Liquid chromatography-high resolution mass spectrometry (LC-HRMS) was performed using a Waters ACQUITY UPLC system, equipped with a binary solvent delivery manager and a sample manager. This system was, coupled with a Waters Micromass Q-TOF Premier Mass Spectrometer, equipped with an electrospray interface (Waters Corporation, Milford, MA).

An Acquity BEH C18 column (100 mm × 2.1 mm; i.d., 1.7 $\mu$m; Waters, Milford, USA) was maintained at 50 °C and eluted with gradient solvent from A : B (95 :) to A : B (0 : 100) at a flow rate of 0.40 mL/min, where A is aqueous formic acid (0.1% (v/v) formic acid) and B is acetonitrile (0.1% (v/v) formic acid) with an injection volume of 5.0 $\mu$L. The following gradient was applied (0–4.00 min: 95.0% A + 5.0% B; 4.00–6.00 min: 80.0% A + 20.0% B; 6.00–8.00 min: 75.0% A + 25.0% B; 8–12.5 min: 50.0% A + 50.0% B; 12.5–13.5 min: 15.0% A + 85.0% B; 13.5–15 min: 0.0% A + 100.0% B).

The MS analyses were performed using positive and negative ions channels. The ionization conditions were optimized, and the operating parameters were as follows: Polarity: positive/ negative; Capillary voltage: 3.0 kV/2.8 kV; Sampling cone: 35 V/45 V; Collision energy: 3 eV/3 eV; Source temperature: 115 °C/115 °C; Desolvation temperature: 350 °C/350 °C; Desolvation gas: 600 L/hr/ 600 L/hr; Scan range: m/z 50–1,500/m/z 50–1,500; Scan time: 0.3 s/0.3 s; Interscan time: 0.02 s/0.02 s.

## Animals and diets

Six-week-old female C57BL/6 mice were purchased from the SLAC Laboratory (Shanghai, China). The animal protocols used in this study were approved by the Shanghai University of Traditional Chinese Medicine (approval number 2014019). The mice were housed under 22 °C–23 °C with a 12 h light/dark cycle. After a 1 week adaptation period, we randomly divided the seven-week-old mice into three groups. They were fed a chow diet (Chow, 10% of calories derived from fat, Research Diets; D12450B), an HF diet (HF, 60% of calories derived from fat, Research Diets; D12492), or a diet supplemented with 1% RSE (HF + RSE). The mice had free access to food and water for 8 weeks. We weighed the food intake and body weight every 2 days.

## Rectal temperature measurement

At the end of 8 weeks, the rectal temperature of the mice was recorded three times at 3 PM by using an instrument for measuring rectal temperature, at intervals of 2 days.

## Intraperitoneal glucose tolerance test

All mice were fasted for 12 h overnight at the end of the preventive experiment. For the intraperitoneal glucose tolerance test (IPGTT), we collected the blood samples from the tail vein for determination of baseline glucose values (0 min). Next intraperitoneal injections of glucose (1 g/kg body weight) were administered to all the mice in 15 min, and blood glucose levels were measured at regular intervals (15, 30, 60, and 90 min) after the injection of glucose.

## Intraperitoneal insulin tolerance test

The mice were not fasted for the intraperitoneal insulin tolerance test (IPITT). Similar to the IPGTT, the basal blood glucose levels (0 min) were measured from the tail vein before

the injection of insulin (0.75 U/kg body weight). The insulin was diluted in physiological saline. Next, additional blood glucose levels were measured at 15, 30, 60, 90, and 120 min after the injection of insulin.

## Serum chemistry analysis

The mice were fasted for 10 h overnight at the end of the animal preventive experiment; the next day, all mice were anesthetized using 20% urethane before sample and tissue collection. Blood samples were drawn from the heart using a 1 mL syringe. After clotting at room temperature for over 2 h, the serum was separated from the blood samples. After centrifugation, 120 µL of serum was drawn from every sample, and the serum total cholesterol (TC), triglyceride (TG), low-density lipoprotein cholesterol (LDL-c), high-density lipoprotein cholesterol (HDL-c), alanine aminotransferase (ALT) and aspartate aminotransferase (AST) levels were analyzed using a Hitachi 7020 Automatic Analyzer.

## Morphological analysis of white adipose tissue

To examine the structure of WAT, the WAT samples were fixed in 4% paraformaldehyde. The tissue samples were sectioned at 5 µm intervals and stained with Hematoxylin and Eosin (H & E). The stained samples were examined under a light microscope.

## Reporter assay

The reporter assay was performed using the Dual-Luciferase Reporter Assay System (Promega, USA) as previously described. The expression plasmids for pCMXGal-hPPAR $\alpha$, $\beta$, $\gamma$ and the Gal4 reporter vector MH100 × 4-TK-Luc were co-transfected with a reporter construct so that 1 µg of the relevant plasmid combined with 1µg of reporter plasmids and 0.1 µg of pREP7 (*Renilla luciferase*) reporter could be used to normalize transfection efficiencies. The transfection mixture, which contained 10 µg of total plasmids and 15 µl FuGENE®HD per ml of DMEM, was added to HEK293T cells for 24 h and then removed. The PPAR $\alpha$, $\beta$, $\gamma$ agonists (Fenofibric acid, GW7647, Pioglitazone) and 2.5, 5, 10, 20, 50, 100, 200, 400, 600, 800, 1,000 µg /ml of RSE were added to fresh media and the cells were incubated for another 24 h to determine luciferase activity.

## Quantitative real-time polymerase chain reaction (Real time qPCR)

The total RNA of WAT and BAT was extracted using the RNAiso Plus (Takara, Dalian, China). RNA is unstable, and to facilitate stable long-term preservation, we used the RevertAid First Strand cDNA Synthesis Kit (Thermo Scientific, Wilmington, Delaware, USA) for the first-strand cDNA (42 °C, 1 h; 70 °C, 5min). The gene expression levels were analyzed using quantitative real-time RT-PCR conducted using the ABI StepOnePlus real-time PCR system (Applied Biosystems, USA). The relative primers involved in the experiments are listed in Table 1. β-Actin was considered an internal control to normalize the expression levels of genes. The cDNA was denatured at 95 °C for 10 min followed by 40 cycles of PCR (95 °C for 15 s, 60 °C for 60 s).

## Statistical analysis

Data were analyzed using SPSS 18.0, and the results were presented as mean ± SEM. Differences were considered significant if $P < 0.05$. Statistical analysis included one-way

**Table 1** Sequences of the primers used in real-time PCR of mouse tissue.

| Gene | Forward primer | Reverse primer |
|---|---|---|
| β-Actin | TGTCCACCTTCCAGCAGATGT | AGCTCAGTAACAGTCCGCCTAGA |
| PPARα | AGGCTGTAAGGGCTTCTTTCG | GGCATTTGTTCCGGTTCTTC |
| PPARβ | AGTGACCTGGCGCTCTTCAT | CGCAGAATGGTGTCCTGGAT |
| PPARγ | CGCTGATGCACTGCCTATGA | AGAGGTCCACAGAGCTGATTCC |
| PGC-1α | TGTTCCCGATCACCATATTCC | GGTGTCTGTAGTGGCTTGATTC |
| PGC-1β | GGGTGCGCCTCCAAGTG | TCTACAGACAGAAGATGTTATGTGAACAC |
| aP2 | CATGGCCAAGCCCAACAT | CGCCCAGTTTGAAGGAAATC |
| ACC | GAATCTCCTGGTGACAATGCTTATT | GGTCTTGCTGAGTTGGGTTAGCT |
| ACO | CAGCACTGGTCTCCGTCATG | CTCCGGACTACCATCCAAGATG |
| UCP1 | CATCACCACCCTGGCAAAA | AGCTGATTTGCCTCTGAATGC |
| UCP2 | GGGCACTGCAAGCATGTGTA | TCAGATTCCTGGGCAAGTCACT |
| UCP3 | TGGCCCAACATCACAAGAAA | TCCAGCAACTTCTCCTTGATGA |
| CD36 | GCTTGCAACTGTCAGCACAT | GCCTTGCTGTAGCCAAGAAC |
| Glut4 | GTAACTTCATTGTCGGCATGG | AGCTGAGATCTGGTCAAACG |

**Table 2** Chemical constituents of RSE identified through liquid chromatography-high resolution mass spectrometry.

| Peak | Rt(min) | MS-Mol. wt.+H | Actual Mol. wt.+H | Formula | Constituents |
|---|---|---|---|---|---|
| 1 | 3.30 | 435.1387 | 435.1359 | $C_{20}H_{22}N_2O_9$ | Glucodichotomine B |
| 2 | 3.362 | 273.0850 | 273.0875 | $C_{14}H_{12}N_2O_4$ | Dichotomine B |
| 3 | 4.070 | 257.0919 | 257.0926 | $C_{14}H_{12}N_2O_3$ | β-carboline alkaloid |
| 4 | 7.59 | 384.1195 | 384.1151 | $C_{19}H_{17}N_3O_6$ | Dichotomine H |
| 5 | 9.238 | 369.1425 | 369.1450 | $C_{20}H_{20}N_2O_5$ | Dichotomine L |
| 6 | 7.614 | 254.0919 | 254.0930 | $C_{14}H_{11}N_3O_2$ | Stellarine A |
| 7 | 8.190 | 338.1138 | 338.1141 | $C_{18}H_{15}N_3O_4$ | Stellarine B |
| 8 | 8.940 | 269.0920 | 269.0926 | $C_{15}H_{12}N_2O_3$ | Stellarine C |
| 9 | 3.793 | 153.0562 | 153.0552 | $C_8H_8O_3$ | Vanillin |
| 10 | 2.058 | 127.0392 | 127.0395 | $C_6H_6O_3$ | 5-Hydroxymethylfurfural |

Notes.
Rt, retention time (min); MS-Mol, wt.+H: primary mass spectrometry; Actual Mol, wt.+H: actual molecular weight.

analysis of variance, the Student's $t$-test, the Kruskal–Wallis H Test, and repeated measures analysis of variance.

# RESULTS

## LC–HRMS detection of the main chemical contituents

We performed the LC–HRMS assay to characterize the constituents in the extract. Ten compounds were putatively identified in the extract according to a previous report (*Chen et al., 2010*). We detected some β-carboline alkaloids, such as stellarine A–C, dichotomine B, H, and L, glucodichotomine B as well as vanillin, 5-hydroxymethylfurfural (Fig. 1 and Table 2).

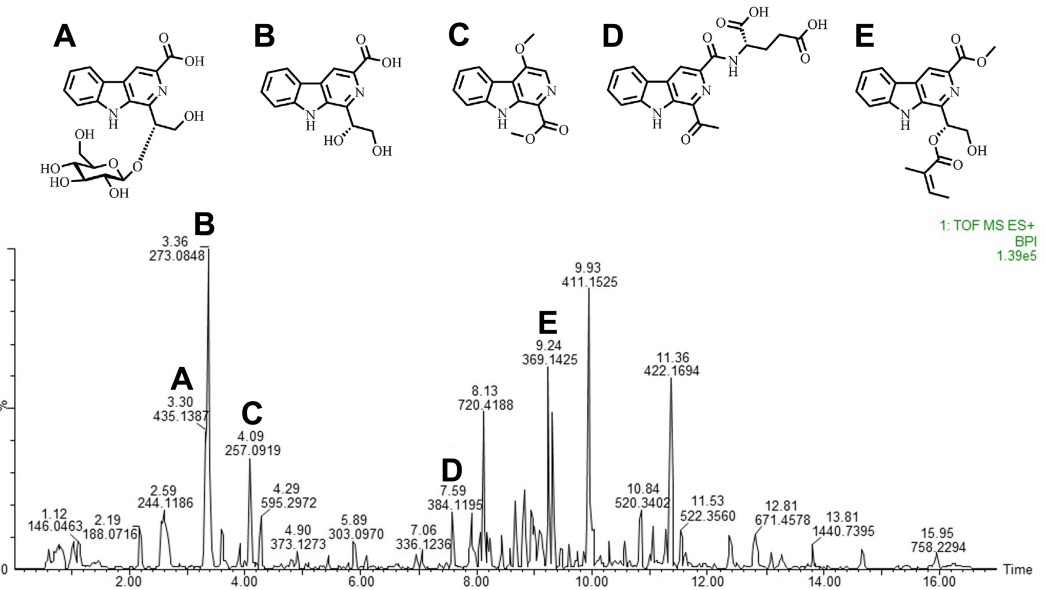

**Figure 1** **Carbolines of RSE identified through liquid chromatography-high resolution mass spectrometry.** Total ion chromatogram of the chemical composition in RSE identified through LC-HRMS, performed using positive ions channel. (A) Glucodichotomine B (B) Dichotomine B (C) $\beta$-carboline alkaloid (D) Dichotomine H (E) Dichotomine L.

## RSE inhibited body weight gain in C57BL/6 mice induced by high-fat diet

To investigate the effect of RSE on body weight gain, we selected the most widely used inbred strain C57BL/6 mouse, which is susceptible to diet-induced obesity, type 2 diabetes and atherosclerosis (*Sun et al., 2016*; *Xia et al., 2016*). The dose of RSE was determined by using a series of complex mathematical operation according to the dose of humans (15 g natural plant medicine/60 kg body weight/day) and pharmacology of traditional Chinese medicine (*Zhang, 2002*). The C57BL/6 mice were fed on a chow diet, HF diet or HF diet supplemented with 1% RSE for 8 weeks. The data revealed that the average body weight in the HF group was significantly higher than that of the Chow group (Fig. 2A), whereas the body weight of the HF + RSE group was evidently lower than that in the HF group from Week 2 to Week 8 (Fig. 2A). The data indicated that RSE could inhibit body weight gain induced by HF-diet in mice. There was no significant difference of food intake between the HF and HF + RSE groups (Fig. 2B). Furthermore, we observed the weekly food intake of the mice. The weekly intake of the three groups remained in a state of equilibrium (Fig. 2C). Thus, the lower body weight in RSE-treated mice was not caused by a lower calorie intake.

Next, we measured the adipocyte size using the H & E stain. The results revealed that the size of WAT in the HF group was considerably larger than that of the Chow group, and RSE treatment reduced the size of WAT in HF + RSE group (Figs. 2D–2E). The data supported the conclusion that RSE inhibits body weight gain.

Inhibition of lipid absorption in the intestine or increase of energy expenditure may result in weight reduction. To test whether RSE affected the lipid absorption and energy expenditure, we determined the total cholesterol and triglyceride content of the feces of the

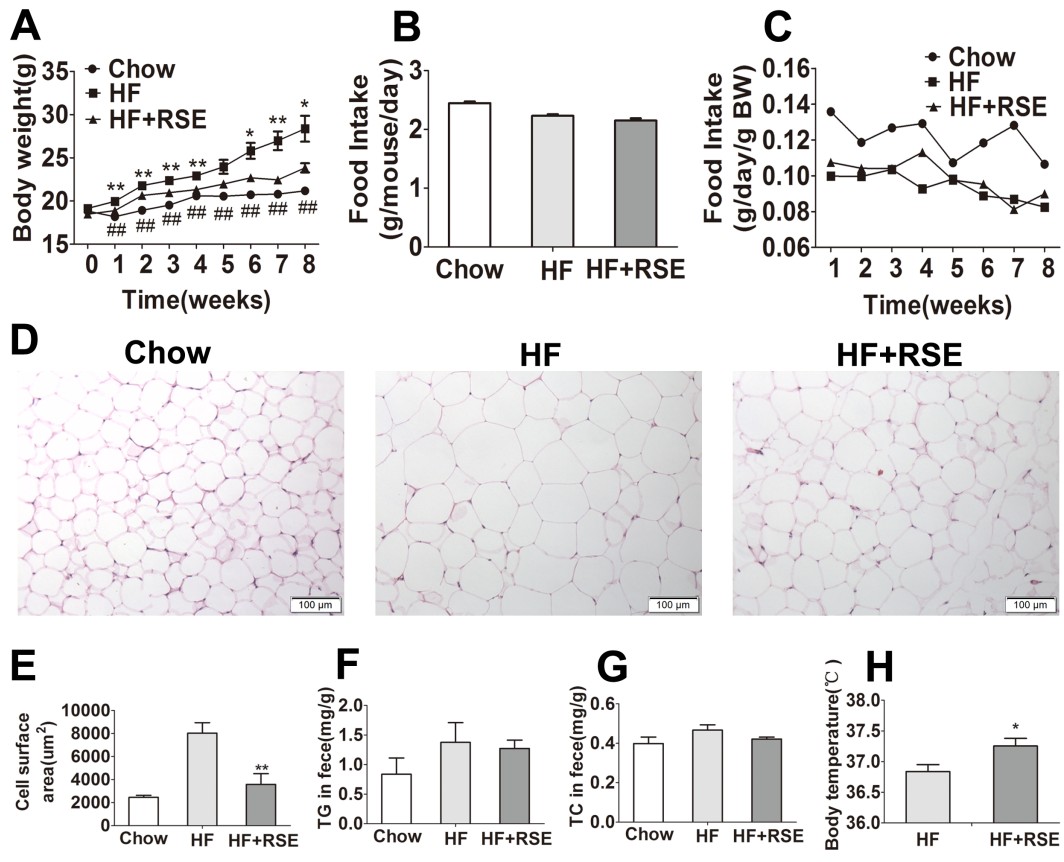

**Figure 2** **RSE prevents metabolic disorders in high-fat-diet -induced obese C57BL/6 mice.** The mice were fed with Chow, HF-diet, and HF-diet mixed with 1% (w/w) RSE for 8 weeks. (A) Body weight (B) Food intake amount (C) Food intake dynamic figure (D) H&E staining of WAT sections (200×) (E) Cell surface area of WAT (F) Feces TG levels (G) Feces TC levels (H) Body temperature. Data are presented as mean ± SEM (Chow: $n = 10$; others: $n = 8$). $*P < 0.05$, $**P < 0.01$ vs the HF group. $\#P < 0.05$, $\#\#P < 0.01$ were the Chow vs the HF group.

mice, and measured rectal temperature. The RSE-treated mice did not exhibit an increase in the TC and TG content of feces (Figs. 2F–2G); however, the body temperature of the RSE-treated mice was notably higher than that of the mice in the other groups (Fig. 2H). Hence, an increase in energy consumption, and not the inhibition of intestinal lipid absorption, may be responsible for the reduction in body weight in HF + RSE group mice.

## RSE reduced fasting blood glucose and ameliorated insulin tolerance in high-fat diet-induced obese C57BL/6 mice at 15 and 30 min

Obesity is a cause for insulin resistance and type II diabetes. Therefore, we measured the fasting blood glucose levels and glucose tolerance in the mice (Fig. 3A). RSE- treated mice exhibited lower fasting glucose levels than did the HF-fed mice (Fig. 3B). However, the blood glucose levels did not change following intraperitoneal injection of glucose. Then we tested the insulin tolerance in the mice. The results revealed that the blood glucose levels of the HF + RSE group were lowered than those of the HF group at 15, and 30 min (Fig. 3C).

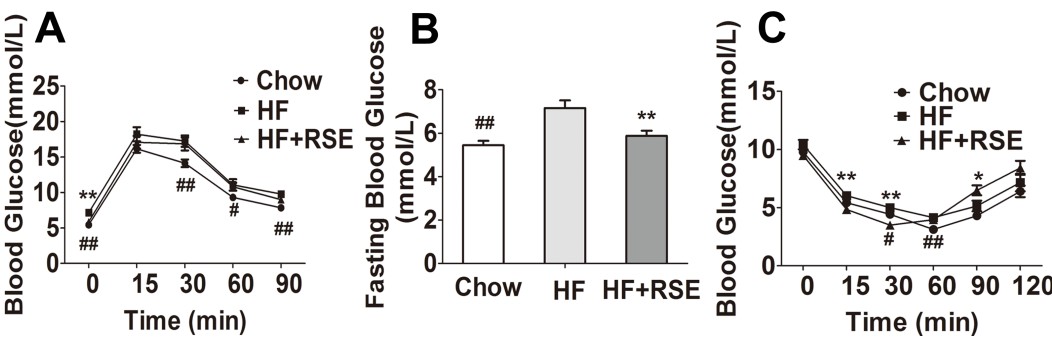

**Figure 3** **RSE improves glucose metabolism and insulin tolerance in high-fat-diet-induced C57BL/6 mice.** (A) Intraperitoneal glucose tolerance test at 0, 15, 30, 60, and 90 min. The mice were fasted for 12 h before measuring blood glucose levels at 0 min (B) Fasting glucose levels (C) Intraperitoneal insulin tolerance test at 0, 15, 30, 60, 90, and 120 min. The mice were not fasted. Data are presented as mean ± SEM (Chow: $n = 10$; others: $n = 8$). *$P < 0.05$, **$P < 0.01$ vs the HF group. #$P < 0.05$, ##$P < 0.01$ were the Chow vs the HF group.

## RSE lowered the lipid profile in serum and liver

Obesity may be accompanied by hyperlipidemia. Therefore, we measured the lipid levels in serum and liver tissue. The fasting serum TG, TC, and LDL-c levels of the HF + RSE group were slightly lower than those of the HF group, although the levels were not significantly different between both groups (Fig. 4A). Similarly, the hepatic TG and TC levels were also lower in the RSE-treated mice (Fig. 4B). Then, we tested the serum levels of ALT and AST, the two indicators of liver damage. The levels of ALT of the HF group were evidently higher than those of the Chow diet-fed mice, indicating potential damage to liver function. RSE treatment, however, did not change the ALT and AST concentrations in HF group mice (Figs. 4C and 4D).

## RSE induced the expression of metabolic gene *in vivo*

The genes for uncoupling proteins (UCPs), namely UCP1, UCP2, and UCP3, are closely related to energy metabolism. In view of the rise of body temperature, we tested the expression of these genes in BAT, which participates in energy consumption and heat production. The expression of the genes in the RSE-treated mice BAT was not notably different from that of the HF-fed mice (Fig. 5A). White beige fat, indicated by high UCP1 expression (*Nedergaard & Cannon, 2014*), which increases energy metabolism, may also be a mechanism of fat reduction. Therefore, we analyzed the expression of UCPs in the WAT. The data revealed that the mRNA of UCP1 and UCP3 increased markedly, suggesting the induction of WAT browning by RSE (Fig. 5B).

PPARs are the ligand-activated nuclear transcription factors regulating the gene expression of glucose and lipid metabolism. A reporter assay was performed to test whether the RSE alters transactivities of PPAR α, β, γ. The results showed that RSE did not change the transcription activity of PPARs, suggesting that RSE does not activate PPARs directly.

We examined the mRNA expression levels of PPARs and their target genes in the WAT. RSE clearly increased the mRNA expression of PPARs and acetyl coenzyme A carboxylase (ACC), acyl-CoA oxidase (ACO), adipose fatty acid-binding protein (aP2), cluster of

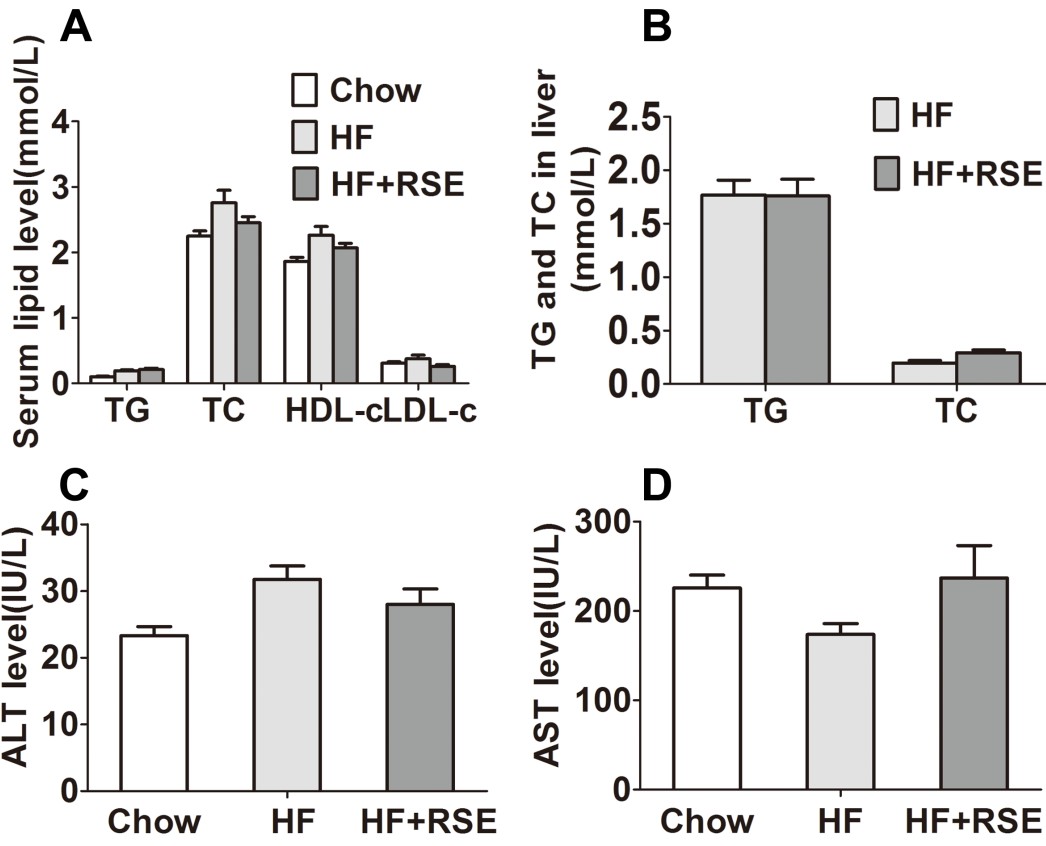

**Figure 4** **Effects of RSE on serum and liver lipid levels in high-fat-diet-induced C57BL/6 mice.** (A) Serum TC, TG, HDL-c, LDL-c levels (B) Liver TG and TC levels (C) ALT levels in serum (D) AST levels in serum. Data are presented as mean ± SEM (Chow: $n = 10$; others: $n = 8$), figure B comparison between group pairs were K–W $H$ test because the data didn't follow normal distribution. $*P < 0.05$, $**P < 0.01$ vs the HF group.

differentiation 36 (CD36), peroxisome proliferator-activated receptor coactivator-1 α and −1 β (PGC-1 α and PGC-1 β), glucose transporter 4 (GLUT 4) as well as UCP1, and UCP3. Among them, PPAR β, PPAR γ, UCP1, UCP3, ACO, aP2, and CD 36 were significant (Fig. 5B). Taken together, the data suggested that RSE may regulate body weight and blood glucose levels through the enhancement of PPAR signaling.

## DISCUSSION

MS is a complex health problem involving several complications and is prevalent in both developed and developing countries. Host genetic and environmental factors can result in MS (*Lim et al., 2016*). *Wu et al. (2016)* reported that physical activity can serve as an effective means to prevent metabolic syndrome. However, physical activity alone does not effectively prevent MS; drugs are also required. Therefore, the development of new drugs to manage MS is necessary.

RS, a common Chinese herbal medicine, has been used in the treatment of deficiency-heat syndrome. Some compounds extracted from RS have different potential

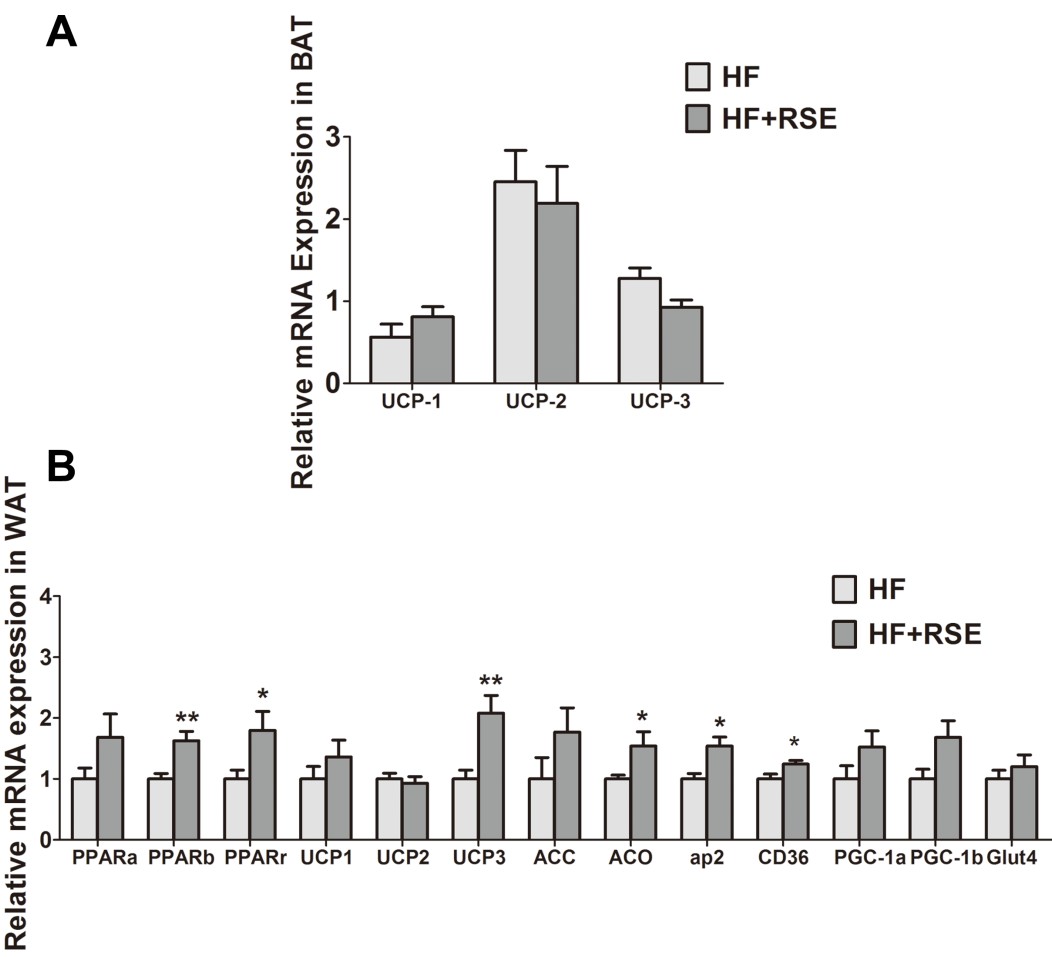

**Figure 5  RSE induced the expression of UCPs, PPARs and their target genes.** (A) Relative expression levels of UCPs in the brown adipose tissue (B) Relative expression levels of UCPs and PPARs target genes in the white adipose tissue. β-actin was used as an internal control for modifying the mRNA level. Data are presented as mean ± SEM (Chow: $n = 10$; others: $n = 8$). *$P < 0.05$, **$P < 0.01$ vs the HF group.

pharmacological effects, for instance, saikosaponin C can prevent Alzheimer's disease in various neuronal models and saikosaponin D can inhibit selectin-mediated cell adhesion (*Jang et al., 2014*). Furthermore, the plant extract of RS is used as a reducing agent to convert gold ions to gold nanoparticles in a biofabrication process. An increasing number applications of RS have been recently discovered, such as prevention of depression and anxiety-like behaviors in rats exposed to repeated restraint stress. However, there was no reported that RS could treat obesity. We should investigated promising new features of Chinese traditional medicine.

Through LC-HRMS, we detected some chemical constituents of RSE that were mainly β-carboline alkaloids ( βCAs)dichotomine B, dichotomine H, dichotomine L, stellarine A–C and glucodichotomine B. βCAs are a type of heterocyclic amines; they are considered to be products of cooking meat (*Lavita et al., 2016*), and widely distributed in the nature and their action is similar to that of indole alkaloids. Recent years, cyclopeptides have received considerable attention from pharmacologists, chemists and biochemists owing to

their various bioactivities such as antiviral, antineoplastic, immunomodulate properties. For example, as a main chemical constituent of Psammosilene tunicoides, stallarine A, a new cyclic heptapeptide (*Zhao et al., 1995*) showed the bacteriostatic activity to a certain extent (*Wang et al., 2012*). Besides, a study reported that glucodichotomine B and neolignan glycosides isolated from the root of RS showed antiallergic activities (*Morikawa et al., 2004*). In a word, βCAs exhibit anti-tumor, anti-microbial, anti-viral (*Li et al., 2006*), anti-oxidative (*Hadjaz et al., 2011*) and insecticidal activities. Moreover, according to some previous studies, we found that many alkaloids were identified to have PPARs agonistic activity: picrasidine C and picrasidine N (*Zhao et al., 2016a*; *Zhao et al., 2016b*), isolated from the root of *Picrasma quassioides*, were identified to have PPARα and PPARβ agonistic activity respectively (*Zhao et al., 2016a*; *Zhao et al., 2016b*); evodiamine, an indole alkaloid extracted from the Chinese medicine evodia, has been shown to inhibit tumor cell invasion and protect the cardiovascular system through activating PPARγ (*Ge et al., 2015*). In the present study, βCAs constituted a large proportion of the RSE; therefore, we suspect that they played a role in treating obesity through regulating PPARs.

In the present study, RSE treatment significantly reduced body weight and WAT size in the C57BL/6 mice compared with HF group. Weight-reduction therapy involves three major methods: reducing food intake, increasing energy expenditure, and inhibiting lipid absorption. We did not observe significant changes in food consumption and lipid absorption in the intestine. Interestingly, the rectal temperature of RSE-treated mice was markedly higher than that of HF-diet-fed mice. These data suggest that RSE could block body-weight gain by increasing the energy metabolism rather than by reducing calorie intake or inhibition of intestinal lipid absorption. Our findings indicated that RS also could be used to increase the body temperature in obese subjects.

RSE-treated mice exhibited lower fasting blood glucose levels and improved insulin tolerance than the mice in the other groups. Obesity is a crucial risk factor for metabolic disorders, moreover, weight reduction may improve insulin tolerance and diabetes. Therefore, the body-weight reduction might contributed to the glucose-lowering effects of RSE.

For the mechanism of increasing energy expenditure, we focused on the mRNA expression levels of related genes in BAT and WAT, which play a vital role in energy metabolism. Recently, "beige adipocytes" have been identified (*Wu et al., 2012*), which share common morphology and function with classical BAT, but they are observed in WAT (*Shin et al., 2016*). Beige adipocytes can promote browning in WAT and increase the expression levels of UCPs, and efficiently increasing energy expenditure by elevating thermogenesis. Therefore, "beige adipocytes" provide a platform for anti-obesity therapy (*Harms & Seale, 2013*; *Pfeifer & Hoffmann, 2015*). Our results suggested that RSE might block body weight gain induced by HF-diet in C57BL/6 mice through the elevation of energy metabolism genes expression promoting the white fat beige, evidenced by the high-level expression of UCPs (*Nedergaard & Cannon, 2014*).

According to the experimental results, the expression of PPARs and downstream genes were increased, indicating that RSE may activate PPARs signaling. Uncoupling protein3 (UCP3) is a mitochondrial anion carrier protein, regarded as an obesity candidate gene. It is mainly distributed in the skeletal muscles and BAT, and it is also expressed in WAT.

UCP3 could mediate the oxidation process and ADP uncoupling phosphorylation process, thereby preventing energy storage in the form of ATP but releasing it in the form of heat. Evidence supported the role of UCP3 in the lipid metabolic, glucose metabolic (*Busiello, Savarese & Lombardi, 2015*), and energy balance of the body, specifically glucose oxidation and insulin sensitivity (*Bezaire, Seifert & Harper, 2007*; *Busiello, Savarese & Lombardi, 2015*). RSE sharply increased the mRNA levels of PPARs and target gene UCP3, thereby suggesting that RS prevents HF-diet-induced obesity in C57BL/6 mice mainly through the activation of PPARs and UCP3 signaling.

## CONCLUSION

RS may alleviate metabolic disorders, by inhibiting body weight increase, reducing fasting blood glucose levels, and ameliorating insulin tolerance in HF diet-induced obese C57BL/6 mice through the increase of UCP3 and PPARs. Our data suggest that RS may be used to prevent metabolic disorders in addition to its traditional uses. However, the potential effects of RS have yet to be discovered, and the identification of active ingredients and elucidation of mechanisms underlying the alleviation of metabolic disorders call for further inquiry.

## ACKNOWLEDGEMENTS

The authors would like to thank F Li (State Key Laboratory of Natural Medicines, Department of Natural Medicinal Chemistry, China Pharmaceutical University, People's Republic of China) for his useful suggestions and revision.

### Funding

The authors received no funding for this work.

### Competing Interests

The authors declare there are no competing interests.

### Author Contributions

- Yin Li performed the experiments, analyzed the data, wrote the paper, prepared figures and/or tables, reviewed drafts of the paper.
- Xin Liu and Yu Fan contributed reagents/materials/analysis tools.
- Baican Yang conceived and designed the experiments.
- Cheng Huang conceived and designed the experiments, reviewed drafts of the paper.

### Animal Ethics

The following information was supplied relating to ethical approvals (i.e., approving body and any reference numbers):

Animal experiments were approved by the Shanghai University of Traditional Chinese Medicine, approval number is 2014019.

## Data Availability

The raw data has been supplied as Data S1.

## Supplemental Information

Supplemental information for this article can be found online at http://dx.doi.org/10.7717/peerj.3305#supplemental-information.

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
