# Peer review of "Radix Stellariae extract prevents high-fat-diet-induced obesity in C57BL/6 mice by accelerating energy metabolism"

_PeerJ, doi:10.7717/peerj.3305_

## Round 0.1 · original submission · Major Revisions

Dear Authors:
Please carefully response to the comments of the reviewers point by point.

Reviewer 1 ·

Basic reporting

It is an interesting study. The authors reported that Radix Stellariae extract could modulate energy metabolism in HFD-mice through UPC and PPAR signaling.

Experimental design

The experimental design minor concerns:

1, the levels of UPC and PPARs proteins should be assayed;
2, the transcriptional activities of PPARs should be assayed.

Validity of the findings

If the compounds' effects on UPC and PPARs could be assayed, it will be better.

Additional comments

The writing should be optimizing, such as the titles of legends. The sentences should be more clear and exact. For example:

Figure 1 Carbolines of RSE identified through liquid chromatography-high resolution mass spectrometry.

Figure 2 RSE prevents metabolic disorders in high-fat-diet (HFD)-induced obese C57BL/6 mice.

Figure 3 RSE improves glucose metabolism and insulin tolerance in HFD-fed C57BL/6 mice.

·

Basic reporting

Your introduction needs more detail. I suggest that you improve the description at lines 97- 99 and to provide more justification for your study (specifically, you should provide references for your statement).

Experimental design

no comment

Validity of the findings

no comment

Reviewer 3 ·

Basic reporting

Here Li and co-workers show a nice set of experiments performed with root extracts of Stellaria dichotoma, using a murine model of induced obesity in C57BL/6 strain under different diets. With minimal exceptions, the article is well written in professional English and perfectly understandable for any reader whose mother tongue is not English.
They cited a total of 31 references, which this reviewer considere adequate, even though some of them should be updated (e.g. ref. Desvergne and Wahli, 1999 about PPAR and refs. From Samec et al about UCP inter-talking between tissues should be updated). Some additional references are required at specific points in the Discussion section (see below)
Figures are professional-grade and they show clearly the data. From the scientific point of view, they do not require any improvement. However, in order to make them more easily legible, line plots final size should be increased because it is difficult to read the symbols (e.g. Fig 2C, Fig 3A and C).
Results obtained are compelling and mostly support the hypothesis presented, i.e. Radix Stellariae partially prevent some metabolic parameters associated with high-fat diet-induced obesity in the murine model mentioned above. However, there are some lacks that authors should address and make some additional comments. Furthermore, some of the conclusions made should be reconsidered or maybe explained conveniently throughout the Discussion.

Experimental design

According to the authors, all the experimental protocols performed here have passed the corresponding ‘Ethical Committee’ at the Shanghai University
The only lack in the experimental design is the absence of a group of animals receiving a normal diet plus the RS extract, which would give additional information about the potential use of this extract in normal conditions, without any basal effects on lipid metabolism under a normal diet. In addition, it would give us information about the potential effects of the RS on fasting glucose in animals taking the normal chow diet.
Authors mentioned in the M&M section the use fo ANOVA and Student’s t tests but also the use fo the K-W H test, which is a non-parametric way to compare data not following normal distribution or arranged in range groups. However, authors do not mentioned in the figure legends when they did use this test and why they chose it. This point should be clarified.
As a minor point, authors should mention at what time of the day temperature was registered, as this parameter is highly influenced by a circadian rhythm.
The rest of the experimental design appear to be complete, and authors have made a nice range of different technical approaches in this study.

Validity of the findings

Authors show compelling results to conclude about the partial effectiveness of RS in reducing the metabolic syndrome consequences. However, some of the findings were not discussed while others have been considered too optimistically.
• Authors have convincingly demonstrated the presence of at least 10 compounds within the extract, using a highly sensitive technique. However, there is a lack of reference(s) regarding many of the aspects of these compounds in the Discussion. Furthermore authors missed to interpret and discussed the potential actions of any of these compounds in the context of lipid metabolism and/or PPAR regulators (As examples, please see to Zhao et al 2016, J Nat Prod; Hadjaz et al, 2011, Eur J Med Chem)
• Authors cannot infer ‘regulation by RSE’ from the data. It is convincingly demonstrated that RSE induce the expression of some mediators, but this might be an indirect rather than a directly-mediated action. Authors should be cautious with the expression ‘RSE regulates’, because other types of cell culture experiments are required for demonstrating these actions.
• Results regarding RSE preventing insulin resistance should also be taken cautiously. Even though RSE intake clearly (significantly) reduces fasting glucose in HF-diet mice, the results obtained in the insulin experiment are not so clear (Fig 3C) and actually authors did not mention that ‘HF+RSE’ group shows even higher glucose blood levels after 90 or 120 min of insulin treatment. Anyway, differences are not great enough and also in the case of glucose tolerance, no differences between HF or HF+RSE were observed.
• Authors did not discuss anything about the inversion observed in the ALT and AST in the HF diet groups

Additional comments

In the present manuscript Li et al have shown interesting data about the effect of radix Stellariae extracts as a preventing effect of the metabolic syndrome in obesity, using a murine model in a HF diet. Although the data and experiments appear technically correct, there are some gaps in the interpretation of data that require further attention. The raw data sent by authors are correctly labeled and easy to follow up and reveals the work done. In this regard, the only lack within the data is the units of the qPCR studies.
Beside some slight changes in a couple of figures, the major issue would be the inclusion of a normal chow diet with the extract and also the interpretation and conclussions made from some of the data included (please refer to the specific comment above). Otherwise the results seem to be compelling and demonstrate the effects of RSE in this model and it potential use.

Reviewer 4 ·

Basic reporting

(1)The introduction needs to be more organized. I suggest that the statement of purpose of current study (lines 89-92, 99- 101) be put at the end of the introduction part
(2)In this manuscripts there are several faults in grammar, punctuation and sentence stuctureI (in line 80, 83,217,295, 318 ).
(3)Besides, abbreviation of “high fat diet” is mixed throughout the artical and you should unify to make it clear.

Experimental design

(1)Please explain why you choose the very dose, 1% of RSE. How did you determine this concentration?
(2)Why do you use rectal temperature as the index of energy expenditure?

Validity of the findings

(1)An explanation of blood glucose levels in RSE group were markedly lowered at 15, 30, and 90 min, while not at 60 min or 120 min should be provided.
(2)You compared related indices in RSE group to HF group, not to control group, why?
(3)The partial results on the lipid profile in serum and liver didn’t support the subtitle “RSE improved the lipid profile in serum and liver” conclusively. You can change the subtitle or explain why you use the word “improved”.
.

Additional comments

The statement of the article is clear and concise. The research questions are well defined, thus the results are meaningful to fill the knowledge gap. Why choose the C57BL/6 mice instead of the normal one to perform the experiment?

Reviewer 5 ·

Basic reporting

No comment

Experimental design

No comment

Validity of the findings

No comment

Additional comments

The purpose of this study was to investigate the effects of Radix Stellariae extract in metabolism disorders. The authors demonstrated that RS inhibited body weight increase, reduced fasting glucose levels, and ameliorated insulin resistance in HFD-induced obese C57BL/6 mice through the activation of UCP3 and PPARs. Then, they conclude that RS. The results obtained are of interest and could have important clinical application. The study design is adequate to address the scientific questions raised in the paper and the figures are adequate.

---

## Round 0.2 · Minor Revisions

Dear authors:

Your responses to reviewers, if necessary, should be included in the manuscript not only in the responsive letter. For example, how to decide the dose and why to select the specific mouse strain. Please coordinate them in the manuscript!

# # Additional note from staff: You have used the phrase "The development of traditional Chinese medicine will not only effectively prevent the side effects of western medicine, but also conserve our traditional Chinese medicine." however, as currently worded that is an unsubstantiated (and overly broad) claim. While you may have demonstrated the efficacy of an active ingredient used in traditional remedies, it is not appropriate to then generalise to the benefits of all Chinese Medicines. Therefore you should re-word or remove this statement. # #

---

## Round 0.3 · accepted · Accept

The quality of the revised version is acceptable for publication